

# Salivary microbial changes during the first 6 months of orthodontic treatment

Mei Zhao[1], Min Liu[1], Wei Chen[1], Haiping Zhang[2], Yuxing Bai[2] and Wen Ren[1]

[1] Department of Preventive Dentistry, School of Stomatology, Capital Medical University, Beijing, China
[2] Department of Orthodontics, School of Stomatology, Capital Medical University, Beijing, China

## ABSTRACT

**Background**. Orthodontic treatment is widely used to treat malocclusion. However, the influence of treatment on the oral microbiome remains unclear. In this study, we investigated salivary microbial changes in patients undergoing orthodontic treatment.
**Methods**. In total, 19 orthodontic patients participated in this study. Saliva samples were collected at the following three timepoints: before (T0) and 3 months (T1) and 6 months (T2) after the placement of orthodontic appliances. High-throughput sequencing was performed based on the 16S rRNA gene V4 region.
**Results**. The phyla of Proteobacteria, Bacteroidetes, Firmicutes, Actinobacteria and Fusobacteria were predominant. Observed Species, Chao1 and ACE, which represent $\alpha$ diversity, were significantly decreased at T1 and subsequently increased at T2. In addition, the $\beta$ diversity at T1 based on the Bray-Curtis distances differed from T0 and T2. The relative abundances of *Prevotella*, *Porphyromonas* and *Peptostreptococcus* were decreased with treatment, whereas those of *Capnocytophaga* and *Neisseria* exhibited the opposite results. In total, 385 of 410 operational taxonomic units were shared at T0, T1 and T2. The co-occurrence networks with hub nodes at T1 were the most complex.
**Conclusion**. Orthodontic treatment temporarily affected the saliva microbial community. This dynamic alteration in species did not induce deterioration in oral health. Oral hygiene instructions were necessary and should be emphasized during each visit. Further studies with longer observation periods and more participants are required.

# INTRODUCTION

Orthodontic treatment is an effective method for correcting malocclusion, which is a common disease in the oral and maxillofacial region. The increased accumulation of dental plaque with the placement of orthodontic appliances induces the occurrence of dental diseases (*Lee et al., 2005*; *Van Gastel et al., 2008*). Dental plaque is the primary etiological factor of white spot lesions, caries, gingivitis and periodontitis, which represent complications of orthodontic treatment and pose a great threat to oral health (*Al-Anezi, 2014*). In some cases, severe periodontitis even causes the failure of orthodontic treatment. Therefore, long-term plaque control should be performed throughout the entire treatment process. It is recommended that maintaining good oral hygiene and receiving regular periodontal examinations are priorities for orthodontic patients, especially during the early stage of treatment.

Corresponding author
Wen Ren, renwen90@foxmail.com

Previous studies have shown that microbial differences are often detected at specific sites, such as supragingival plaque in dental caries and gingivitis and subgingival plaque in periodontitis (*Abusleme et al., 2013*; *Xu et al., 2014*). In addition to dental plaque, saliva, which is regarded as a microbial repository and transport medium, is widely studied in dental research. Various studies have indicated that saliva is affected by oral health status and the quantity and types of microbes during orthodontic treatment. *Jing et al., (2019)* detected carious pathogens in orthodontic patients and reported that *Streptococcus mutans* abundance remained stable during the first 6 months and significantly increased at 18 months. Meanwhile, *Bergamo et al., (2019)* focused on the periodontal complex and found that green, yellow and orange complexes decreased at 3 months .

Many reports have evaluated orthodontic treatment from a microbiological aspect using traditional methods, such as culture, DNA-DNA hybridization, and polymerase chain reaction. The oral cavity harbors more than 700 species; of these species, 35% cannot be cultivated under current experimental conditions (*Dewhirst et al., 2010*). High-throughput sequencing overcomes the limitations of traditional methods, providing a more comprehensive perspective of the global microbial community. This technique has been widely used in gut and dental research (*Citronberg et al., 2018*; *Serena et al., 2018*; *Teng et al., 2015*; *Xu et al., 2015*). In this study, we examined a longitudinal cohort of orthodontic patients over a 6-month period. Salivary microbial communities were characterized and compared to better understand the microbial changes occurring during the early stage of orthodontic treatment.

## MATERIAL AND METHODS

### Ethics statement

This study was approved by the Ethics Committee of Capital Medical University School of Stomatology (CMUSH-IRB-KJ-PJ-2017-04). Written informed consent was obtained from all the participants or their guardians.

### Study design and sample collection

In total, 70 participants from the Department of Orthodontics underwent an oral examination. Nineteen of these patients underwent a six-month follow-up and were recruited for this study. During the follow-up, the following three timepoints were selected: before the start of the orthodontic treatment (T0) and 3 months (T1) and 6 months (T2) after the treatment. None of the participants had severe periodontal diseases, colds, or systemic diseases or received antimicrobial or anti-inflammatory therapy within 6 months prior to the study. Saliva samples were collected by a saliva collection kit (Zeesan Biotech Co., Xiamen, China). The patients were not allowed to eat, drink, smoke or chew gum for 30 min before giving their samples. In total, 2 ml of unstimulated saliva was collected, and saliva stabilization solution was added to the samples. All samples were stored at −80 °C until DNA extraction. The Quigley-Hein Plaque Index was selected and evaluated by one experienced dentist. Oral hygiene instructions were provided during each appointment.

**Table 1 Basic information.**

|  | Plaque index | Raw reads | Clean reads | OTU | Good's coverage (%) |
|---|---|---|---|---|---|
| T0 | 2.37 ± 0.40 | 78785 ± 15280 | 75167 ± 13922 | 239 ± 40 | 99.94 ± 0.01 |
| T1 | 1.84 ± 0.30 | 74011 ± 13588 | 69250 ± 7047 | 211 ± 57 | 99.93 ± 0.02 |
| T2 | 1.79 ± 0.32 | 72738 ± 7396 | 70064 ± 6777 | 229 ± 27 | 99.93 ± 0.02 |

## DNA extraction, PCR amplification and high-throughput sequencing

Genomic DNA was extracted using prepIT · L2P (DNA Genotek, Canada) according to the manufacturer's instructions. The DNA purity was determined using a NanoDrop 8000 Spectrophotometer (Thermo, Waltham, MA, USA). The hypervariable region V4 of the bacterial 16S rRNA gene was amplified via PCR using the primers 515F (5′-GTGCCAGCMGCCGCGGTAA-3′) and 806R (5′-GGACTACHVGGGTWTCTAAT-3′). The PCR amplicons were sequenced using the Illumina HiSeq platform with a pair-end 250-bp strategy at the BGI Institute (Shenzhen, China). The sequences were submitted to the NCBI Sequence Read Archive under accession number PRJNA648248.

## Bioinformatic analysis and statistical analysis

The 16S rRNA gene sequences were processed using the QIIME pipeline (ver. 1.9.1). The sequencing data were demultiplexed using a unique barcode assigned to each sample. The sequences were trimmed when the average quality score over a 25-bp sliding window was less than 20. The pair-end sequences were joined by FLASH (ver. 1.2.11) and then clustered into operational taxonomic units (OTUs) at a 3% dissimilarity cut-off using USEARCH (ver. 7.0.1090). The taxonomic information was generated using Greengenes (ver. 13.5), with a threshold of 0.6.

The rarefaction curves, $\alpha$ diversity (Observed Species, Chao1, and ACE), $\beta$ diversity (Bray-Curtis distance), and OTU and taxonomic relative abundance were calculated by QIIME. The Friedman test and post hoc comparisons were used to compare the $\alpha$ diversity and differential OTUs; ANOSIM was used to compare the $\beta$ diversity. A Venn diagram was constructed to describe the core and unique microbiome. Spearman's rank correlation coefficients were calculated between each pair of OTUs. The co-occurrence network was generated and analyzed using Cytoscape (ver. 3.2.1). All statistical tests were performed using R software (ver. 3.5.1); $p$-values <0.05 were considered statistically significant.

# RESULTS

## Basic information of the participants and sequencing

To explore the changes in the salivary microbiota during orthodontic treatment, we performed a 6-month longitudinal study including the following three timepoints: before (T0) and 3 months (T1) and 6 months (T2) after treatment. In total, 19 participants were recruited. The average age of the participants was 21.1 ± 7.4 years; the Friedman test and $\chi^2$ analysis revealed no significant differences in terms of plaque index or gender (Table 1). The bacterial community profile in each sample was determined by 16S rRNA gene amplification targeting the V4 region using the Illumina HiSeq platform. After processing,
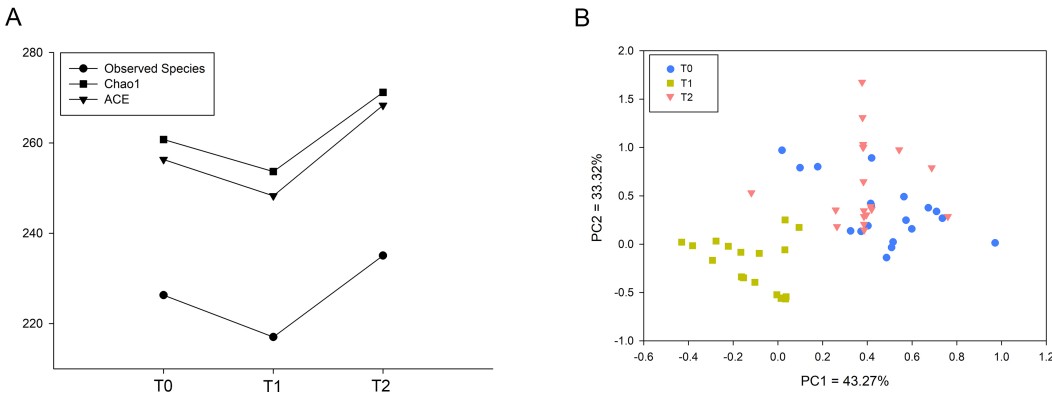

**Figure 1 Comparison of microbial diversity of T0, T1 and T2.** (A) Sequences (60,518) were randomly subsampled to obtain equal numbers of sequences from each dataset. Indices, namely, Observed Species, Chao1 and ACE, representing $\alpha$ diversity are shown. (B) Phylogenetic distances between samples were calculated via Bray–Curtis distance. Each dot in the scatter plot represents one sample. The percentage of variation is indicated on the $x$ and $y$-axes. T1 significantly differed from T0 and T2 (ANOSIM, $p < 0.05$).

we obtained a total of 4,075,131 clean reads, ranging from 60,518 to 90,158 per sample (Table 1). Good's coverage was greater than 99.9% in all the samples. Good's coverage and the rarefaction curves indicated that the sequencing coverage was adequate under the chosen depth (Table 1 & Fig. S1). These sequences were clustered into 410 OTUs using a 3% dissimilarity cut-off.

## Microbial diversity in the microbial community

The indices Observed Species, Chao1 and ACE representing microbial richness were calculated from 60,518 reads. These three indices were significantly decreased at T1 and increased at T2 (Fig. 1A). To investigate the microbial structure, we calculated the $\beta$ diversity based on the Bray-Curtis distance and visualized the microbial structure by principle coordinate analysis (PCoA) plots. The analysis revealed that the T1 samples tended to cluster together and were well separated from the T0 and T2 samples (Fig. 1B).

## OTUs and taxa with different abundances

In total, 13 phyla, 21 classes, 32 orders, 55 families, and 78 genera were observed in this study. Five phyla, Proteobacteria, Bacteroidetes, Firmicutes, Actinobacteria and Fusobacteria, were predominant in all three groups, accounting for more than 99% of all bacteria. To further explore the changes in specific taxa throughout the first six months of orthodontic treatment, we compared the relative abundance of the salivary microbiota (Fig. 2, Fig. S2). The abundance of *Prevotella*, including OTU368, OTU118, OTU085, OTU277, and OTU325, decreased as the treatment progressed (Fig. 2A, Fig. S2E). *Porphyromonas* (OTU034), which is a genus of *Bacteroidetes*, and *Peptostreptococcus* (OTU253) exhibited similar results, whereas *Capnocytophaga* (OTU047) and *Neisseria* (OTU305) exhibited the opposite results (Figs. 2A–C, Fig. S2E).

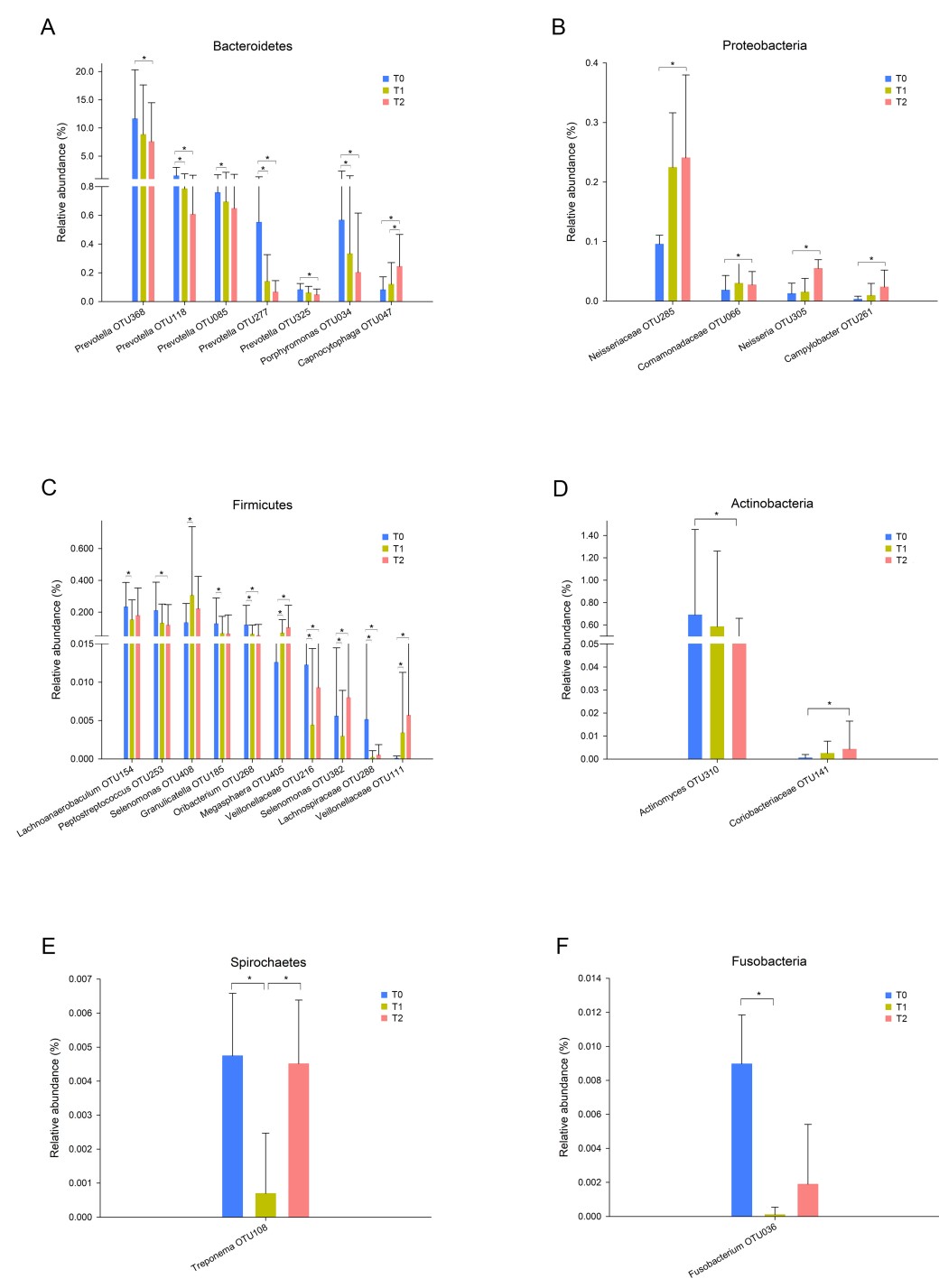

**Figure 2 Taxa with different relative abundances at the OTU level.** (A) Bacteroidetes, (B) Proteobacteria, (C) Firmicutes, (D) Actinobacteria, (E) Spirochaetes, (F) Fusobacteria. Bars represent the mean relative abundance ($\pm$SD). $^*p < 0.05$, Friedman test.

## Core microbiome and co-occurrence networks

Despite the OTUs of different relative abundances, 385 of the 410 OTUs were shared among all three groups (Fig. 3). The remaining 25 OTUs are also shown in Fig. 3, and
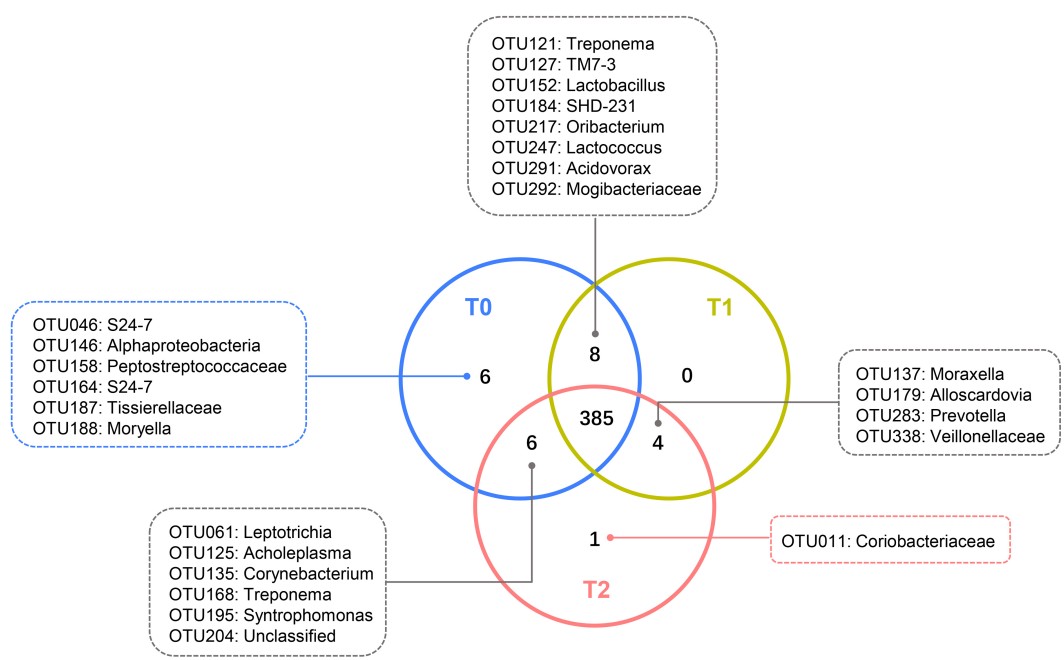

**Figure 3 Shared and unique OTUs of T0, T1 and T2.** The Venn diagram shows the number of each oral bacterial community. A total of 385 OTUs were shared by all the three groups. OTUs in the box were unique OTUs.

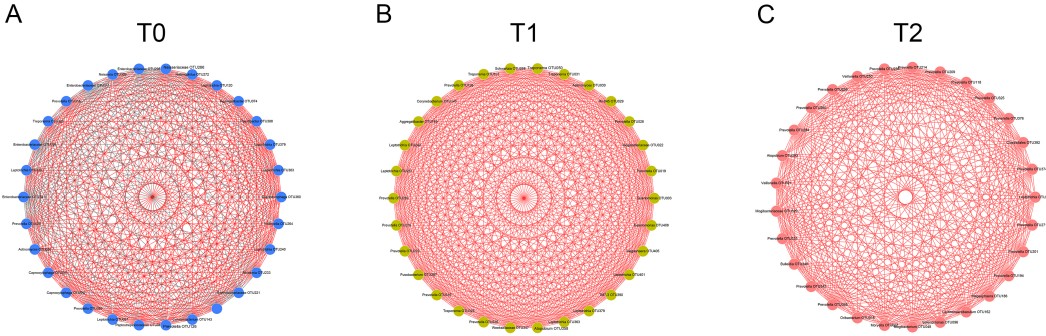

**Figure 4 Co-occurrence networks of hub nodes at T0 (A), T1 (B) and T2 (C).** Spearman's correlation coefficients (SCC) were calculated between different OTUs. Each node represents one OTU. The edge between two nodes represents significant correlations ($|SCC| \geq 0.6, p < 0.05$). Red and gray edges indicate positive and negative correlations, respectively.

the relative abundances of these OTUs were less than 0.01% (Tables S1-1). The core microbiome was defined as the OTUs present in all 57 samples in this study. In total, 26 OTUs were retained. *Neisseria* (OTU278), *Prevotella* (OTU368) and *Streptococcus* (OTU192) predominated among these 26 OTUs, with relative abundances of 9.47%, 9.34% and 6.82%, respectively (Table S1-2).

To detect the relationships of the microorganisms, co-occurrence networks were generated. By calculating and visualizing the Spearman's correlation coefficients for all

OTUs in the T0 group, we initially obtained a network of 393 OTUs (nodes) with 4,611 linkages (edges). Nodes with less than 15 edges were removed. The remaining 30 nodes were identified as hub nodes, with 408 edges (289 positive edges and 119 negative edges, Fig. 4A). Similarly, the networks of the T1 and T2 groups were generated and had no negative edges (Figs. 4B, 4C). Of all three networks, the T1 network exhibited the most linkages ($p < 0.05$). *Prevotella* and *Leptotrichia* were the most prevalent genera (Table S2).

## DISCUSSION

Increasing numbers of people choose to receive orthodontic treatment. The complications of orthodontic treatment are worthy of attention. Previous studies have indicated that the placement of orthodontic appliances increased the risk of dental caries and periodontal disease, which are the most common diseases of the oral cavity. The etiology and progression of caries or periodontitis can be better understood by focusing on the consortia of organisms rather than one or several species (*Xu et al., 2015*). Thus, in this study, we used 16S rRNA gene high-throughput sequencing to explore the microbial changes in orthodontic patients to provide a general view of the bacterial community. Consistent with previous studies, the phyla Proteobacteria, Bacteroidetes, Firmicutes, Actinobacteria and Fusobacteria were the most abundant (*Wang et al., 2019*). Our results indicate that the microbial richness was reduced at T1 and increased at T2 (Fig. 1A). In addition, the community structure at T1 clustered apart from that at T0 and T2 (Fig. 1B). However, the cross-sectional studies conducted by Sun et al. and Wang et al. suggest an increased Shannon index in orthodontic patients compared with that in controls, indicating greater microbial diversity (*Sun et al., 2018*; *Wang et al., 2019*). Sample sources, ages, eating habits, frequency of brushing teeth, and type of appliances are all factors that can affect the results (*Dogramaci, Naini & Brennan, 2020*). Synergistic, mutualistic and antagonistic effects among microorganisms contribute to the development of polymicrobial interactions. Given these complex interactions, we generated co-occurrence networks at T0, T1 and T2 and selected hub nodes (Fig. 4). The networks at T1 differed from those at T0 and T2, with the most linkages, indicating that the microbial community structure is temporary and not permanent.

*Neisseria* is an early colonizer of the tooth surface, and some studies have indicated that *Neisseria* is associated with improved oral health or reduced gingivitis (*Lif Holgerson et al., 2020*). Previous studies reported that the abundance of *Neisseria* was reduced over time following treatment with fixed appliances (*Koopman et al., 2015*; *Wang et al., 2019*). In contrast, our study revealed the opposite result (Fig. 2B). The oral health conditions were potentially improved because we provided regular oral hygiene instructions before the initiation of treatment and reinforced this information during each visit. *Prevotella* spp. are gram-negative, nonmotile and rod-shaped bacteria that thrive under anaerobic growth conditions (*Moller et al., 2020*). *Prevotella* spp. can colonize the human mouth and are considered host-associated bacteria. Several *Prevotella* species are opportunistic pathogens and known for their role in periodontal disease (*Li et al., 2020*). One systematic review concluded that the subgingival pathogen levels exhibited temporary increases after orthodontic appliance placement and that these levels returned to the pretreatment

levels after several months (*Guo et al., 2017*). This finding indicates that orthodontic treatment might not permanently induce periodontal disease by affecting the subgingival periodontal pathogen level. In contrast, in this study, the relative abundance of *Prevotella* was decreased throughout the first 6 months of orthodontic treatment (Fig. 2A). Other studies have reported similar results (*Wang et al., 2019*). Bergamo et al. reported that *Prevotella nigrescens* was present at high levels before bonding using checkerboard DNA-DNA hybridization and that these levels were decreased 60 and 90 days after orthodontic appliance placement (*Bergamo et al., 2019*). However, not all *Prevotella* species exhibited the same changes. The *Prevotella intermedia* levels exhibited no significant differences during the observation period (*Bergamo et al., 2019*; *Guo et al., 2019*). Another important role of *Prevotella* spp. involves providing key nutrients to *Peptostreptococcus* sp. (*James, 2010*). The fact that the relative abundance of *Peptostreptococcus* (OTU253) exhibited a trend similar to that of *Prevotella* provides evidence supporting this important role (Fig. 2C).

Enamel demineralization is caused by the effects of acid products on the carbohydrate metabolism of bacterial species. These species grow 6 or 12 weeks after orthodontic appliance bonding (*Sanpei, Endo & Shimooka, 2010*). Demineralization can cause white spot lesions and may result in dental caries during orthodontic treatment if not treated. *S. mutans*, *Streptococcus sobrinus* and *Lactobacillus* spp. are the main causative microorganisms of enamel demineralization. Many studies have reported changes in these species using different methods at different timepoints. For example, through the traditional culture method, significantly higher *Lactobacillus* CFU counts were found at 2-month follow-up following fixed orthodontic treatment (*Kupietzky et al., 2005*). A significant increase in *Streptococcus* spp. was noted only 1 week after the start of therapy in Reichardt's study using matrix-assisted laser desorption/ionization time-of-flight mass spectrometry (*Reichardt et al., 2019*). *Jing et al., (2019)* reported that the *S. mutans* abundance remained stable during the first 6 months and was significantly increased at 18 months; *Lactobacillus* abundance exhibited a slight but nonsignificant increase, as demonstrated by qPCR. Using Dentocult SM and Dentocult LB, Maret et al. found that wearing a fixed orthodontic appliance was associated with high levels of *S. mutans* and *Lactobacillus* spp. (*Maret et al., 2014*). However, in our study, the relative abundance and prevalence of *S. mutans*, *S. sobrinus* and *Lactobacillus* spp. did not obviously change before and after 6 months of treatment based on the 16S rRNA gene high-throughput sequencing results. Thus, determining which method is more reliable and the time point at which orthodontic treatment can influence the progression of microbial changes is worthy of further investigation. Notably, *Neisseria* (OTU278), *Prevotella* (OTU368) and *Streptococcus* (OTU192) were the predominant genera in the core microbiome (Table S1-2).

This study suggests that alterations occur in the oral microbiota following orthodontic treatment. The orthodontic treatment temporarily affected the saliva microbial diversity. As the relative abundances of *S. mutans* and *Lactobacillus* spp. showed no significant differences, orthodontic treatment does not induce deterioration in oral health from this perspective. Oral hygiene instructions were necessary and should be emphasized during each visit.

One limitation of this study is that some aspects of malocclusion, such as crowing and crossbite, were not considered. Given the relatively small sample size and short observation time, it is necessary to be aware that our research is a preliminary investigation of the salivary microbial community in orthodontic patients.

## CONCLUSIONS

In this study, we found that the placement of orthodontic appliances may have a temporary impact on the salivary microflora. The dynamic alteration in species and orthodontic treatment did not induce deterioration of oral health. Further studies of high methodological quality involving more participants and longer-term observations are required to provide more reliable evidence regarding this issue.

### Funding
This work was supported by the grant from Beijing Municipal Administration of Hospitals Clinical Medicine Development of Special Funding Support (ZYLX201703). The funders had no role in study design, data collection and analysis, decision to publish, or preparation of the manuscript.

### Grant Disclosures
The following grant information was disclosed by the authors:
Beijing Municipal Administration of Hospitals Clinical Medicine Development of Special Funding Support: ZYLX201703.

### Competing Interests
The authors declare there are no competing interests.

### Author Contributions
- Mei Zhao conceived and designed the experiments, performed the experiments, analyzed the data, prepared figures and/or tables, and approved the final draft.
- Min Liu performed the experiments, authored or reviewed drafts of the paper, and approved the final draft.
- Wei Chen and Haiping Zhang analyzed the data, authored or reviewed drafts of the paper, and approved the final draft.
- Yuxing Bai conceived and designed the experiments, prepared figures and/or tables, and approved the final draft.
- Wen Ren conceived and designed the experiments, analyzed the data, prepared figures and/or tables, and approved the final draft.

### Human Ethics
The following information was supplied relating to ethical approvals (i.e., approving body and any reference numbers):

Capital Medical University School of Stomatology granted Ethical approval to carry out the study within its facilities (Ethical Application Ref: CMUSH-IRB-KJ-YJ-2017-04).

## Data Availability

The 16S rRNA gene sequences are available at NCBI Sequence Read Archive: PRJNA648248.

## Supplemental Information

Supplemental information for this article can be found online at http://dx.doi.org/10.7717/peerj.10446#supplemental-information.

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
