# Peer review of "Salivary microbial changes during the first 6 months of orthodontic treatment"

_PeerJ, doi:10.7717/peerj.10446_

## Round 0.1 · original submission · Major Revisions

Please address the issues raised in the reviews and resubmit.

·

Basic reporting

A. Text
The English language should be improved, with more professional terms. Some examples include:
“Many studies revealed that placement of orthodontic appliances influenced cleansing of the oral cavity, thus inducing the accumulation of dental plaque (Lee et al. 2005; van Gastel et al. 2008).”
“The etiology and progression of caries or periodontitis are closely related to the oral microbiome rather than a single or several species. “(What does it mean?)
“Our results indicated that microbial richness (gender, species or levels?)

B. Intro & background
i. Text: “Other evidence also showed that improper orthodontic force could alter the oral microbial ecosystem and increase the potential for pathogenicity (Ong & Wang 2002).”
My question: Is this topic actually reported by literature in this way? Please check the reference.
ii. In concern about the importance of saliva in diagnosis of microbial environment, the reference citations are not adequate.
Text: “Saliva, which is regarded as a microbial repository and transport medium, is also widely used in dental research in addition to dental plaque. Various studies indicated that saliva was affected by oral health status as well as the quantity and types of microbes during orthodontic treatment”.
My question: In the references, there are only HolgersonL et al. 2020. Jing analysed the salivary levels of S. mutans and Lactobacillus, and Zhao et al 2019 analysed in aligner appliance, what should not be compared to brackets. I think it is necessary comparison with researches that was analysed interaction among microorganisms in the saliva.
iii. The aim is not clear:
Text: “Salivary microbial communities were characterized and compared to better understand microbial changes during the early stage of orthodontic treatment.”
My question: what does “microbial communities” mean? Phyla, gender, species? Or group or complex.

Experimental design

i. The research is within scope of the journal.
ii. The question is relevant, but it is not clear: “Salivary microbial communities were characterized and compared to better understand microbial changes during the early stage of orthodontic treatment”
My question: Did they compare phyla, genders, species? Did the sequencing also evaluate species? The results did not show this clearly. And it is important to support the topics addressed in discussion.
iii. Description of methodology is based, clearly, in the literature and, then, is correct. But some aspects of the malocclusion sample are also very relevant, like crowing, crossbite.
Saliva sample: Time that saliva was collected. The aliquot utilized in the analysis. What kind of buffer solution was utilized?
I think it is necessary to describe this aspect.

Validity of the findings

i. The results are inconclusive.
ii. The statistic is adequate.
iii. The figure 4 is inconclusive.
iv. The results showed data about phyla and genera. The discussion addresses specific concerns about dental caries and periodontal disease, in which some species of bacteria are not investigated. And attention: the salivary levels and in situ levels are different, and the comparison with the literature should be considered this aspect.
Therefore, the discussion could not compare the specific results of this research with those literature results pointed by the authors, specially about aligners appliance.

Additional comments

My conclusion: within the points considered, I think this study is relevant and necessary. The orthodontic research should be pointing out more information about microbial interaction in orthodontic treatment. The study was well planned and done. Some adjustments are required to be coherent with the proposal, results and discussion. The major adjustments are necessary to be published.

Reviewer 2 ·

Basic reporting

In this study, the authors conducted to better understand on orthodontic patients’ salivary microbial changes over a 6-month period orthodontic treatment.

Experimental design

1. Could authors supple the reasons why 6-month period was chosen for observation?

2. The number of participant patients involved in this study was 19 which was not big enough.

3. In method part, hypervariable region V4 of the bacterial 16S rRNA gene was PCR amplified with the specific premiers. Could authors explain how these specific premiers could be applied for various bacterial in salivary samples?

4. In method part, “Rarefaction curves, α diversity (Observed species, Chao1, ACE), β diversity (Bray-Curtis distance), and OTU relative abundance were calculated by QIIME”. Could authors add some brief introductions on each index?

Validity of the findings

1. In discussion part, authors analyzed and compared the alternation of the community structure in different term. However little reasons on how contributing those changes was indicated. Please list some reasonable information.

2. From you results, what clinical significance could be inferred? Please add this information on your discussion part.

Additional comments

In this study, the authors conducted to better understand on orthodontic patients’ salivary microbial changes over a 6-month period orthodontic treatment.

1. Could authors supple the reasons why 6-month period was chosen for observation?

2. The number of participant patients involved in this study was 19 which was not big enough.

3. In method part, hypervariable region V4 of the bacterial 16S rRNA gene was PCR amplified with the specific premiers. Could authors explain how these specific premiers could be applied for various bacterial in salivary samples?

4. In method part, “Rarefaction curves, α diversity (Observed species, Chao1, ACE), β diversity (Bray-Curtis distance), and OTU relative abundance were calculated by QIIME”. Could authors add some brief introductions on each index?

5. In discussion part, authors analyzed and compared the alternation of the community structure in different term. However little reasons on how contributing those changes was indicated. Please list some reasonable information.

6. From you results, what clinical significance could be inferred? Please add this information on your discussion part.

---

## Round 0.2 · accepted · Accept

Thank you for addressing the reviewers' comments.

·

Basic reporting

The English language was improved.

i. "Other evidence also showed that improper orthodontic force could alter the oral microbial ecosystem".
This sentence was deleted.

ii. Concern about the importance of saliva in diagnosis of microbial environment the reference citations are not adequate. “Saliva, which is regarded as a microbial repository and transport medium, is also widely used in dental research in addition to dental plaque. Various studies indicated that saliva was affected by oral health status as well as the quantity and types of microbes during orthodontic treatment”. In the references there are only HolgersonL et al. 2020. Jing analysed the salivary levels in orthodontic patients, and Zhao et al 2019 in aligner appliance that should not comparing with brackets.
They modified this section and added more references focusing on studies of salivary microbial changes following fixed appliance installation.

iii. The aim is not clear: “Salivary microbial communities were characterized and compared to better understand microbial changes during the early stage of orthodontic treatment.” What means “microbial communities “
This question was clarified.

Experimental design

The questions were answered and the text was adapted.

Validity of the findings

The findings were validated.

Additional comments

The requests were met, and the limitations discussed. At that moment, I think it should be accepted for publication.